# GENERATING INTERPRETABLE IMAGES WITH CONTROLLABLE STRUCTURE

**S. Reed, A. van den Oord, N. Kalchbrenner, V. Bapst, M. Botvinick, N. de Freitas**
Google DeepMind
{reedscot,avdnoord,nalk,vbapst,botvinick,nandodefreitas}@google.com

## ABSTRACT

We demonstrate improved text-to-image synthesis with controllable object locations using an extension of Pixel Convolutional Neural Networks (PixelCNN). In addition to conditioning on text, we show how the model can generate images conditioned on part keypoints and segmentation masks. The character-level text encoder and image generation network are jointly trained end-to-end via maximum likelihood. We establish quantitative baselines in terms of text and structure-conditional pixel log-likelihood for three data sets: Caltech-UCSD Birds (CUB), MPII Human Pose (MHP), and Common Objects in Context (MS-COCO).

## 1 INTRODUCTION

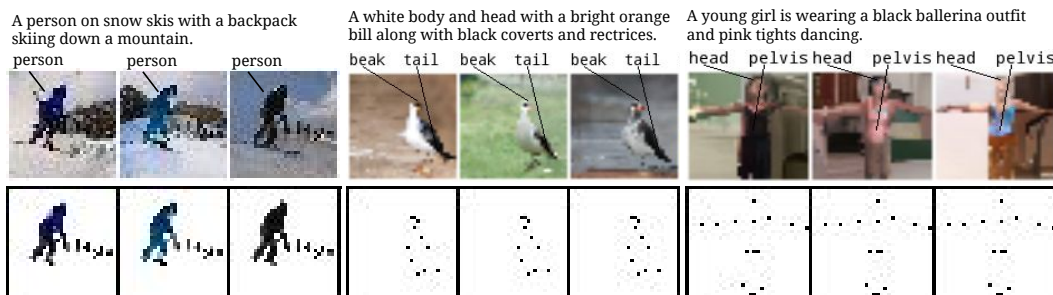

Figure 1: Examples of interpretable and controllable image synthesis. Left: MS-COCO, middle: CUB, right: MHP. Bottom row shows segmentation and keypoint conditioning information.

Image generation has improved dramatically over the last few years. The state-of-the-art images generated by neural networks in 2010, e.g. (Ranzato et al., 2010) were noted for their global structure and sharp boundaries, but were still easily distinguishable from natural images. Although we are far from generating photo-realistic images, the recently proposed image generation models using modern deep networks (van den Oord et al., 2016c; Reed et al., 2016a; Wang & Gupta, 2016; Dinh et al., 2016; Nguyen et al., 2016) can produce higher-quality samples, at times mistakable for real.

Three image generation approaches are dominating the field: generative adversarial networks (Goodfellow et al., 2014; Radford et al., 2015; Chen et al., 2016), variational autoencoders (Kingma & Welling, 2014; Rezende et al., 2014; Gregor et al., 2015) and autoregressive models (Larochelle & Murray, 2011; Theis & Bethge, 2015; van den Oord et al., 2016b;c). Each of these approaches have significant pros and cons, and each remains an important research frontier.

Realistic high-resolution image generation will impact media and communication profoundly. It will also likely lead to new insights and advances in artificial intelligence. Understanding how to control the process of composing new images is at the core of this endeavour.

Researchers have shown that it is possible to control and improve image generation by conditioning on image properties, such as pose, zoom, hue, saturation, brightness and shape (Dosovitskiy et al., 2015; Kulkarni et al., 2015), part of the image (van den Oord et al., 2016b; Pathak et al., 2016), surface normal maps (Wang & Gupta, 2016), and class labels (Mirza & Osindero, 2014; van den Oord et al., 2016c). It is also possible to manipulate images directly using editing tools and learned generative adversarial network (GAN) image models (Zhu et al., 2016).

Language, because of its compositional and combinatorial power, offers an effective way of controlling the generation process. Many recent works study the image to text problem, but only a handful have explored text to image synthesis. Mansimov et al. (2015) applied an extension of the DRAW model of Gregor et al. (2015), followed by a Laplacian Pyramid adversarial network post-processing step (Denton et al., 2015), to generate $32 \times 32$ images using the Microsoft COCO dataset (Lin et al., 2014). They demonstrated that by conditioning on captions while varying a single word in the caption, we can study the effectiveness of the model in generalizing to captions not encountered in the training set. For example, one can replace the word "*yellow*" with "*green*" in the caption "*A yellow school bus parked in a parking lot*" to generate blurry images of green school buses.

Reed et al. (2016a), building on earlier work (Reed et al., 2016b), showed that GANs conditioned on captions and image spatial constraints, such as human joint locations and bird part locations, enabled them to control the process of generating images. In particular, by controlling bounding boxes and key-points, they were able to demonstrate stretching, translation and shrinking of birds. Their results with images of people were less successful. Yan et al. (2016) developed a layered variational autoencoder conditioned on a variety of pre-specified attributes that could generate face images subject to those attribute constraints.

In this paper we propose a gated conditional PixelCNN model (van den Oord et al., 2016c) for generating images from captions and other structure. Pushing this research frontier is important for several reasons. First, it allows us to assess whether auto-regressive models are able to match the GAN results of Reed et al. (2016a). Indeed, this paper will show that our approach with auto-regressive models improves the image samples of people when conditioning on joint locations and captions, and can also condition on segmentation masks. Compared to GANs, training the proposed model is simpler and more stable because it does not require minimax optimization of two deep networks. Moreover, with this approach we can compute the likelihoods of the learned models. Likelihoods offer us a principled and objective measure for assessing the performance of different generative models, and quantifying progress in the field.

Second, by conditioning on segmentations and captions from the Microsoft COCO dataset we demonstrate how to generate more interpretable images from captions. The segmentation masks enable us to visually inspect how well the model is able to generate the parts corresponding to each segment in the image. As in (Reed et al., 2016a), we study compositional image generation on the Caltech-UCSD Birds dataset by conditioning on captions and key-points. In particular, we show that it is possible to control image generation by varying the key-points and by modifying some of the keywords in the caption, and observe the correct change in the sampled images.

## 2 MODEL

### 2.1 BACKGROUND: AUTOREGRESSIVE IMAGE MODELING WITH PIXELCNN

Figure 2 illustrates autoregressive density modeling via masked convolutions, here simplified to the 1D case. At training time, the convolutional network is given the sequence $\mathbf{x}_{1:T}$ as both its input and target. The goal is to learn a density model of the form:

$$p(\mathbf{x}_{1:T}) = \prod_{t=1}^{T} p(x_t | \mathbf{x}_{1:t-1}) \qquad (1)$$

To ensure that the model is causal, that is that the prediction $\hat{x}_t$ does not depend on $x_\tau$ for $\tau \geq t$, while at the same time ensuring that the training is just as efficient as the training of standard convolutional networks, van den Oord

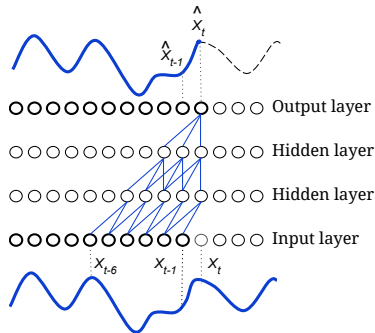

Figure 2: Auto-regressive modelling of a 1D signal with a masked convolutional network.

et al. (2016c) introduce masked convolutions. Figure 2 shows, in blue, the active weights of $5 \times 1$ convolutional filters after multiplying them by masks. The filters connecting the input layer to the first hidden layer are in this case multiplied by the mask $\mathbf{m} = (1, 1, 0, 0, 0)$. Filters in subsequent layers are multiplied by $\mathbf{m} = (1, 1, 1, 0, 0)$ without compromising causality. [1].

---

[1] Obviously, this could be done in the 1D case by shifting the input, as in van den Oord et al. (2016a)

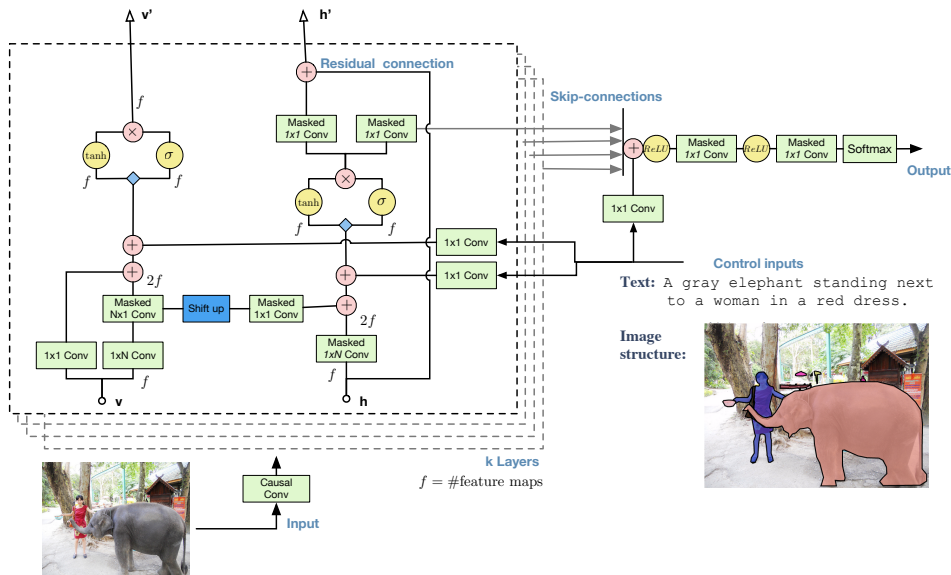

Figure 3: PixelCNN with text and structure conditioning variables.

In our simple 1D example, if $x_t$ is discrete, say $x_t \in \{0, \ldots, 255\}$, we obtain a classification problem, where the conditional density $p(x_t | \mathbf{x}_{1:t-1})$ is learned by minimizing the cross-entropy loss. The depth of the network and size of the convolutional filters determine the receptive field. For example, in Figure 2, the receptive field for $\hat{x}_t$ is $\mathbf{x}_{t-6:t-1}$. In some cases, we may wish to expand the size of the receptive fields by using dilated convolutions (van den Oord et al., 2016a).

van den Oord et al. (2016c) apply masked convolutions to generate colour images. For the input to first hidden layer, the mask is chosen so that only pixels above and to the left of the current pixel can influence its prediction (van den Oord et al., 2016c). For colour images, the masks also ensure that the three color channels are generated by successive conditioning: blue given red and green, green given red, and red given only the pixels above and to the left, of all channels.

The conditional PixelCNN model (Fig. 3) has several convolutional layers, with skip connections so that outputs of each layer layer feed into the penultimate layer before the pixel logits. The input image is first passed through a causal convolutional layer and duplicated into two activation maps, $\mathbf{v}$ and $\mathbf{h}$. These activation maps have the same width and height as the original image, say $N \times N$, but a depth of $f$ instead of 3, as the layer applies $f$ filters to the input. van den Oord et al. (2016c) introduce two stacks of convolutions, vertical and horizontal, to ensure that the predictor of the current pixel has access to all the pixels in rows above; i.e. blind spots are eliminated.

In the vertical stack, a masked $N \times N$ convolution is efficiently implemented with a $1 \times N$ convolution with $f$ filters followed by a masked $N \times 1$ convolution with $2f$ filters. The output activation maps are then sent to the vertical and horizontal stacks. When sending them to the horizontal stack, we must shift the activation maps, by padding with zeros at the bottom and cropping the top row, to ensure that there is no dependency on pixels to the right of the pixel being predicted. Continuing on the vertical stack, we add the result of convolving $2f$ convolutional filters. Note that since the vertical stack is connected to the horizontal stack and hence the ouput via a vertical shift operator, it can afford to look at all pixels in the current row of the pixel being predicted. Finally, the $2f$ activation maps are split into two activations maps of depth $f$ each and passed through a gating tanh-sigmoid nonlinearity (van den Oord et al., 2016c).

The shifted activation maps passed to the horizontal stack are convolved with masked $1 \times 1$ convolutions and added to the activation maps produced by applying a masked $1 \times N$ horizontal convolution to the current input row. As in the vertical stack, we apply gated tanh-sigmoid nonlinearities before sending the output to the pixel predictor via skip-connections. The horizontal stack also uses residual connections (He et al., 2016). Finally, outputs $\mathbf{v}'$ and $\mathbf{h}'$ become the inputs to the next layer.

As shown in Figure 3, the version of model used in this paper also integrates global conditioning information, text and segmentations in this example.

## 2.2 CONDITIONING ON TEXT AND SPATIAL STRUCTURE

To encode location structure in images we arrange the conditioning information into a spatial feature map. For MS-COCO this is already provided by the 80-category class segmentation. For CUB and MHP we convert the list of keypoint coordinates into a binary spatial grid. For both segmentation and the keypoints in spatial format, the first processing layer is a class embedding lookup table, or equivalently a $1 \times 1$ convolution applied to a 1-hot encoding.

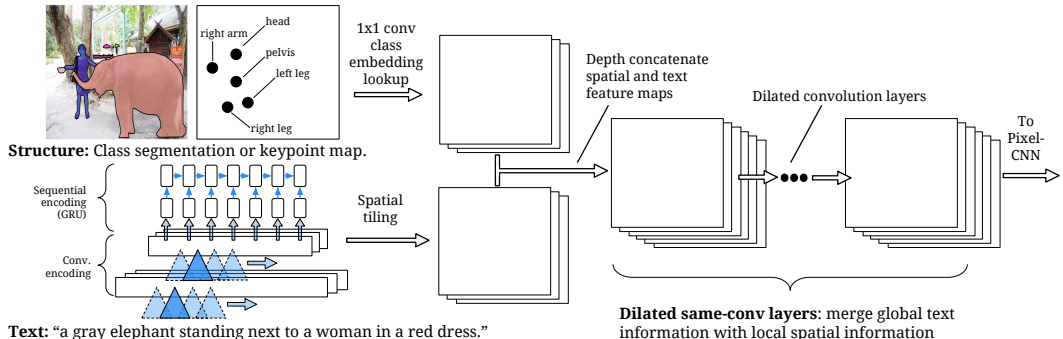

Figure 4: Encoding text and spatial structure for image generation.

The text is first encoded by a character-CNN-GRU as in (Reed et al., 2016a). The averaged embedding (over time dimension) of the top layer is then tiled spatially and concatenated with the location pathway. This concatenation is followed by several layers of dilated convolution. These allow information from all regions at multiple scales in the keypoint or segmentation map to be processed along with the text embedding, while keeping the spatial dimension fixed to the image size, using a much smaller number of layers and parameters compared to using non-dilated convolutions.

## 3 EXPERIMENTS

We trained our model on three image data sets annotated with text and spatial structure.

- The MPII Human Pose dataset (MHP) has around 25K images of humans performing 410 different activities (Andriluka et al., 2014). Each person has up to 17 keypoints. We used the 3 captions per image collected by (Reed et al., 2016a) along with body keypoint annotations. We kept only the images depicting a single person, and cropped the image centered around the person, leaving us 18K images.
- The Caltech-UCSD Birds database (CUB) has 11,788 images in 200 species, with 10 captions per image (Wah et al., 2011). Each bird has up to 15 keypoints.
- MS-COCO (Lin et al., 2014) contains 80K training images annotated with both 5 captions per image and segmentations. There are 80 classes in total. For this work we used class segmentations rather than instance segmentations for simplicity of the model.

Keypoint annotations for CUB and MHP were converted to a spatial format of the same resolution as the image (e.g. $32 \times 32$), with a number of channels equal to the maximum number of visible keypoints. A "1" in row $i$, column $j$, channel $k$ indicates the visibility of part $k$ in entry $(i, j)$ of the image, and "0" indicates that the part is not visible. Instance segmentation masks were re-sized to match the image prior to feeding into the network.

We trained the model on $32 \times 32$ images. The PixelCNN module used 10 layers with 128 feature maps. The text encoder reads character-level input, applying a GRU encoder and average pooling after three convolution layers. Unlike in Reed et al. (2016a), the text encoder is trained end-to-end from scratch for conditional image modeling. We used RMSprop with a learning rate schedule starting at $1e$-4 and decaying to $1e$-5, trained for $200k$ steps with batch size of 128.

In the following sections we demonstrate image generation results conditioned on text and both part keypoints and segmentation masks. Note that some captions in the data contain typos, e.g. "bird is read" instead of "bird is red", and were not introduced by the authors.

## 3.1 TEXT- AND SEGMENTATION-CONDITIONAL IMAGE SYNTHESIS

In this section we present results for MS-COCO, with a model conditioned on the class of object visible in each pixel. We also included a channel for background. Figure 5 shows several conditional samples and the associated annotation masks. The rows below each sample were generated by point-wise multiplying each active channel of the ground-truth segmentation mask by the sampled image. Here we defined "active" as occupying more than 1% of the image.

Each group of four samples uses the same caption and segmentation mask, but the random seed is allowed to vary. The samples tend to be very diverse yet still match the text and structure constraints. Much larger examples are included in the appendix.

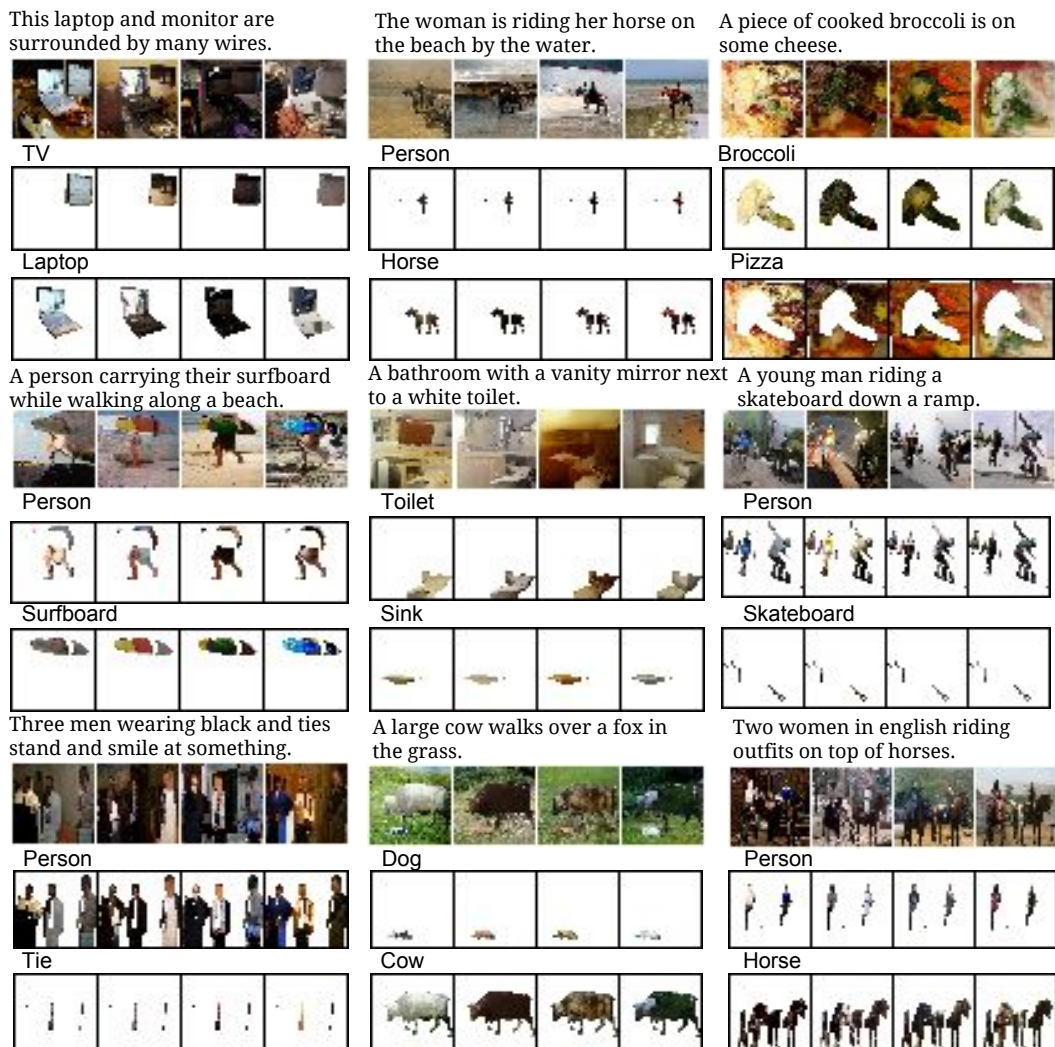

Figure 5: Text- and segmentation-conditional general image samples.

We observed that the model learns to adhere correctly to location constraints; i.e. the sampled images all respected the segmentation boundaries. The model can assign the right color to objects based on the class as well; e.g. green broccoli, white and red pizza, green grass. However, some foreground objects such as human faces appear noisy, and in general we find that object color constraints are not captured as accurately by the model as location constraints.

## 3.2 TEXT- AND KEYPOINT-CONDITIONAL IMAGE SYNTHESIS

In this section we show results on CUB and MHP, using bird and human part annotations. Figure 6 shows the results of six different queries, with four samples each. Within each block of four samples, the text and keypoints are held fixed. The keypoints are projected to 2D for visualization purposes, but note that they are presented to the model as a $32 \times 32 \times K$ tensor, where $K$ is the number of keypoints (17 for MHP and 15 for CUB).

We observe that the model consistently associates keypoints to the apparent body part in the generated image; see "beak" and "tail" labels drawn onto the samples according to the ground-truth location. In this sense the samples are interpretable; we know what the model was meant to depict at salient locations. Also, we observe a large amount of diversity in the background scenes of each query, while pose remains fixed and the bird appearance consistently matches the text.

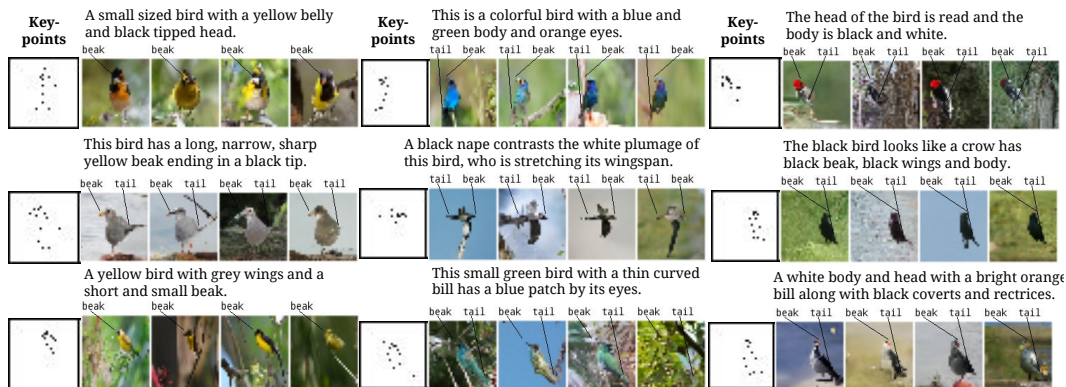

Figure 6: Text- and keypoint-conditional bird samples (CUB data).

Changing the random seed (within each block of four samples), the background details change significantly. In some cases this results in unlikely situations, like a black bird sitting in the sky with wings folded. However, typically the background remains consistent with the bird's pose, e.g. including a branch for the bird to stand on.

Figure 7 shows the same protocol applied to human images in the MHP dataset. This setting is probably the most difficult, because the training set size is much smaller than MS-COCO, but the variety of poses and settings is much greater than in CUB. In most cases we see that the generated person matches the keypoints well, and the setting is consistent with the caption, e.g. in a pool, outdoors or on a bike. However, producing the right color of specific parts, or generating objects associated to a person (e.g. bike) remain a challenge.

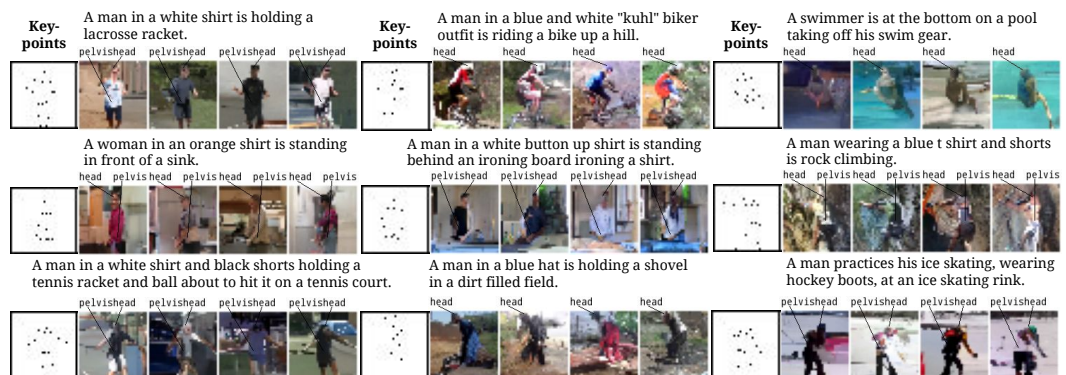

Figure 7: Text- and keypoint-conditional samples of images with humans (MHP data).

We found it useful to adjust the temperature of the softmax during sampling. The probability of drawing value $k$ for a pixel with probabilities $p$ is $p_k^T / \sum_i p_i^T$, where $T$ is the inverse temperature.

| Dataset | train nll | validation nll | test nll |
|---------|-----------|----------------|----------|
| CUB | 2.91 | 2.93 | 2.92 |
| MPII | 2.90 | 2.92 | 2.92 |
| MS-COCO | 3.07 | 3.08 | - |

Table 1: Text- and structure-conditional negative log-likelihoods (nll) in *nats/dim*. Train, validation and test splits include all of the same categories but different images and associated annotations.

Higher values for $T$ makes the distribution more peaked. In practice we observed that larger $T$ resulted in less noisy samples. We used $T = 1.05$ by default.

Ideally, the model should be able to render any combination of of valid keypoints and text description of a bird. This would indicate that the model has "disentangled" location and appearance, and has not just memorized the (caption, keypoint) pairs seen during training. To tease apart the influence of keypoints and text in CUB, in Figure 8 we show the results of both holding the keypoints fixed while varying simple captions and fixing the captions while varying the keypoints.

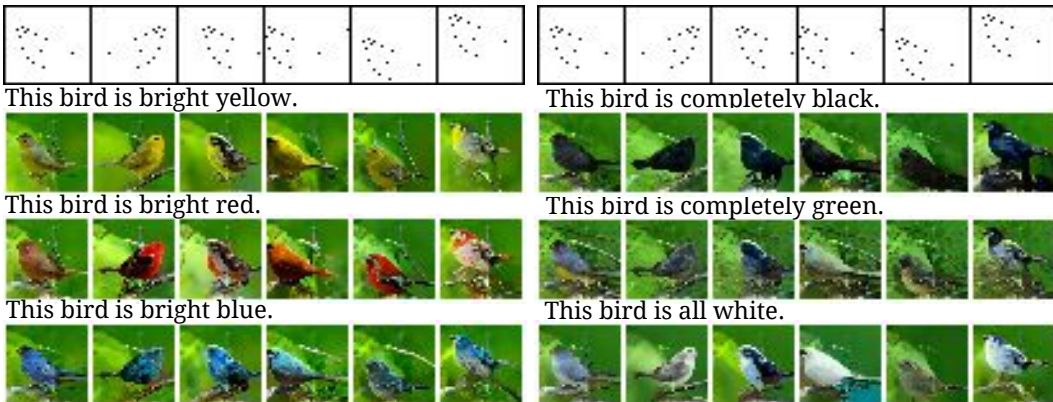

Figure 8: Columns: varying text while fixing pose. Rows (length 6): varying pose while fixing text. Note that the random seed is held fixed in all samples.

To limit variation across captions due to background differences, we re-used the same random seed derived from each pixel's batch, row, column and color coordinates[2]. This causes the first few generated pixels in the upper-left of the image to be very similar across a batch (down columns in Figure 8), resulting in similar backgrounds.

In each column, we observe that the pose of the generated birds satisfies the constraints imposed by the keypoints, and the color changes to match the text. This demonstrates that we can effectively control the pose of the generated birds via the input keypoints, and its color via the captions simultaneously. We also observe a significant diversity of appearance.

However, some colors work better than others, e.g. the "bright yellow" bird matches its caption, but "completely green" and "all white" are less accurate. For example the birds that were supposed to be white are shown with dark wings in several cases. This suggests the model has partially disentangled location and appearance as described in the text, but still not perfectly. One possible explanation is that keypoints are predictive of the category of bird, which is predictive of the appearance (including color) of birds in the dataset.

## 3.3 QUANTITATIVE RESULTS

Table 1 shows quantitative results in terms of the negative log-likelihood of image pixels conditioned on both text and structure, for all three datasets. For MS-COCO, the test negative log-likelihood is not included because the test set does not provide captions.

---

[2]Implemented by calling `np.random.seed((batch, row, col, color))` before sampling.

The quantitative results show that the model does not overfit, suggesting that in future research a useful direction may be to develop higher-capacity models that are still memory- and computationally-efficient to train.

## 3.4 COMPARISON TO PREVIOUS WORKS

Figure 9 compares to MHP results from Reed et al. (2016a). In comparison to the approach advanced in this paper, the samples produced by the Generative Adversarial What-Where Networks are significantly less diverse. Close inspection of the GAN image samples reveals many wavy artifacts, in spite of the conditioning on body-part keypoints. As the bottom row shows, these artifacts can be extreme in some cases.

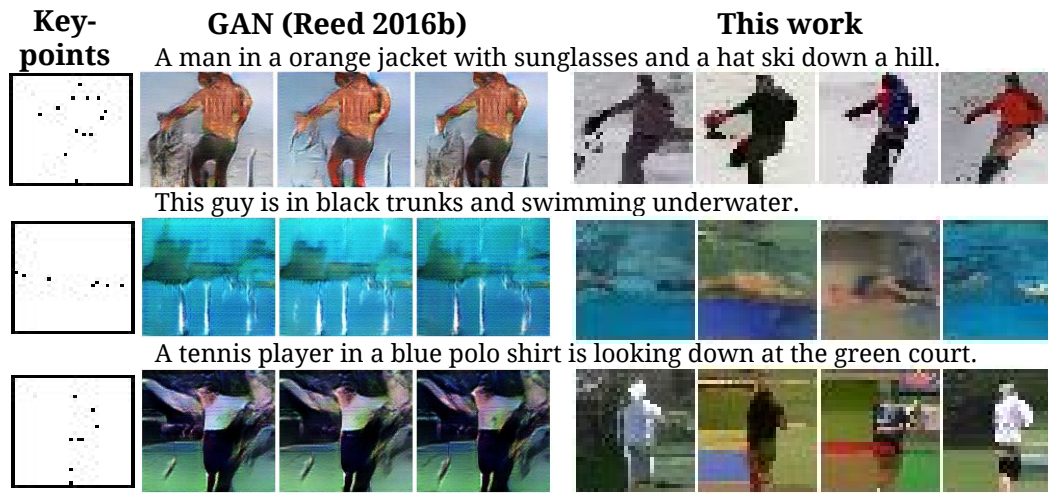

Figure 9: Comparison to Generative Adversarial What-Where Networks (Reed et al., 2016a). GAN samples have very low diversity, whereas our samples are all quite different.

## 4 DISCUSSION

In this paper, we proposed a new extension of PixelCNN that can accommodate both unstructured text and spatially-structured constraints for image synthesis. Our proposed model and the recent Generative Adversarial What-Where Networks both can condition on text and keypoints for image synthesis. However, these two approaches have complementary strengths. Given enough data GANs can quickly learn to generate high-resolution and sharp samples, and are fast enough at inference time for use in interactive applications (Zhu et al., 2016). Our model, since it is an extension of the autoregressive PixelCNN, can directly learn via maximum likelihood. It is very simple, fast and robust to train, and provides principled and meaningful progress benchmarks in terms of likelihood.

We advanced the idea of conditioning on segmentations to improve both control and interpretability of the image samples. A possible direction for future work is to learn generative models of segmentation masks to guide subsequent image sampling. Finally, our results have demonstrated the ability of our model to perform controlled combinatorial image generation via manipulation of the input text and spatial constraints.

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

## 5 APPENDIX

this laptop and monitor are surrounded by many wires

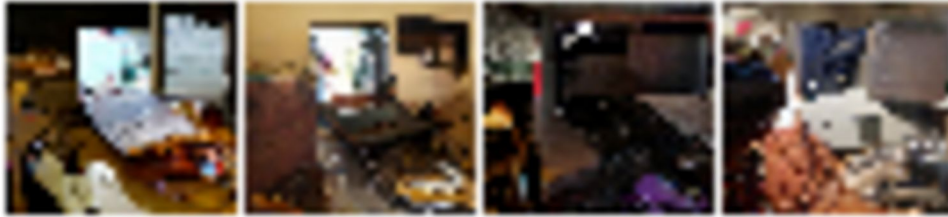

a couple of people standing in a field playing with a frisbee

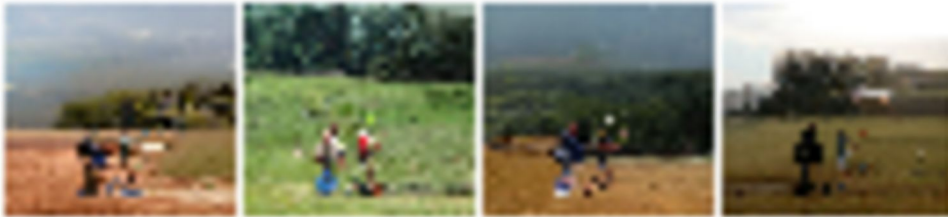

a cat is drinking water from the toilet bowl

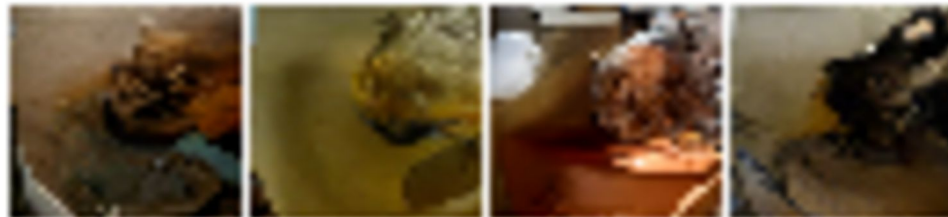

a small child touches the trunk of an elephant standing behind an enclosure

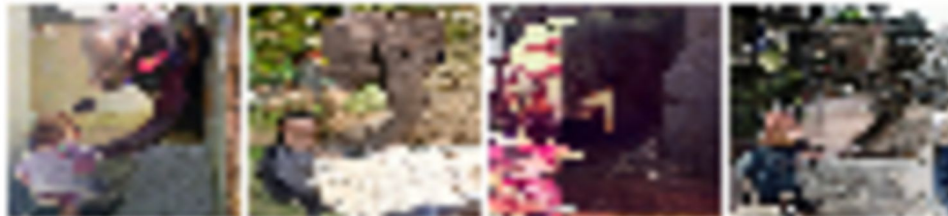

the woman is riding her horse on the beach by the water

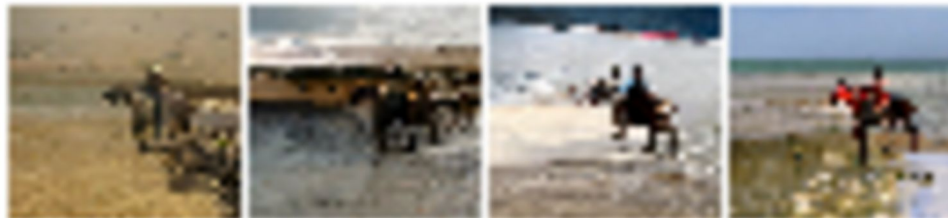

a man in a tie shaking a woman by the hand

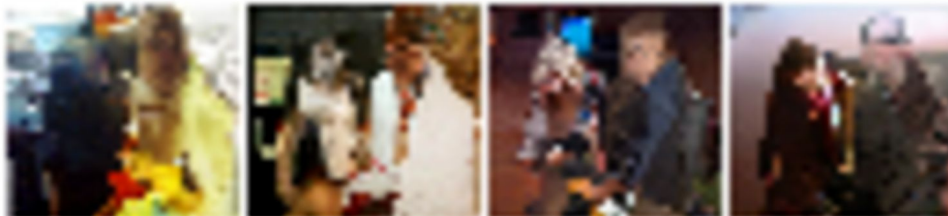

this bird has a long, narrow, sharp yellow beak ending in a black tip.

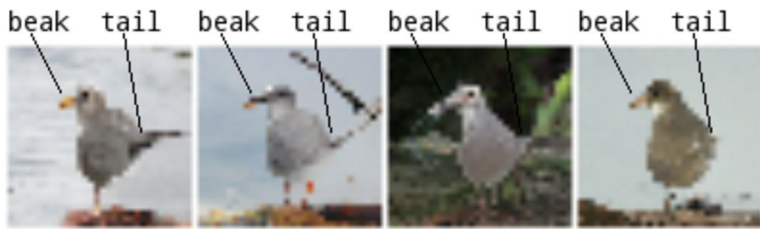

a small sized bird with a yellow belly and black tipped head

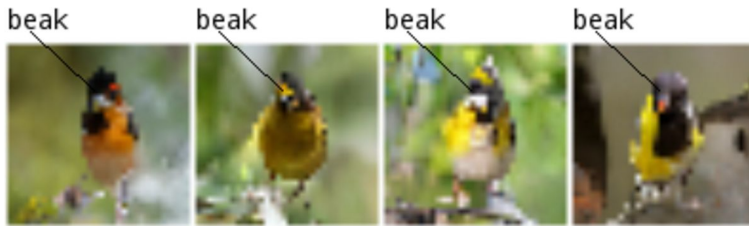

the head of the bird is read and the body is black and white.

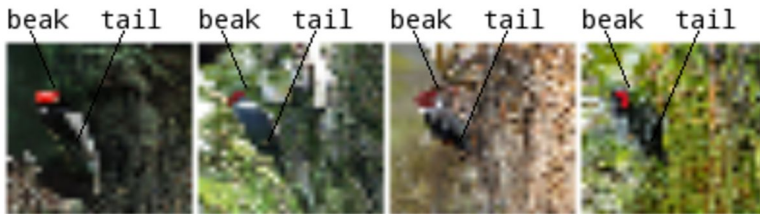

this is a small bird with a white breast and brown wings.

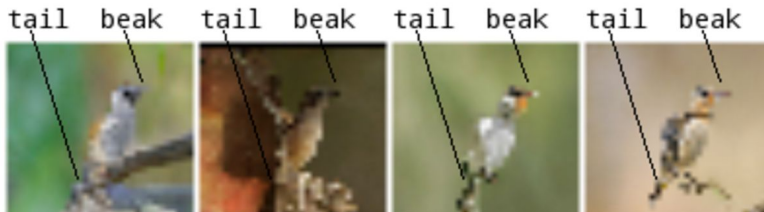

a small multi colored bird with white, brown, and black feathers.

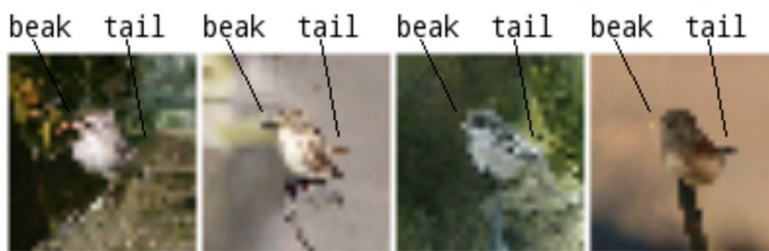

this is a colorful bird with a blue and green body and orange eyes.

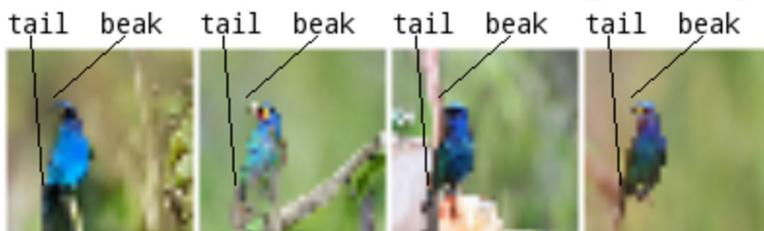

a man in a blue shirt is throwing a brown football on a field.

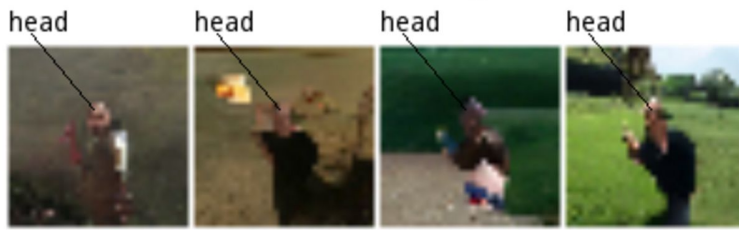

the lady with the blue jacket placed her hands around her shoulder

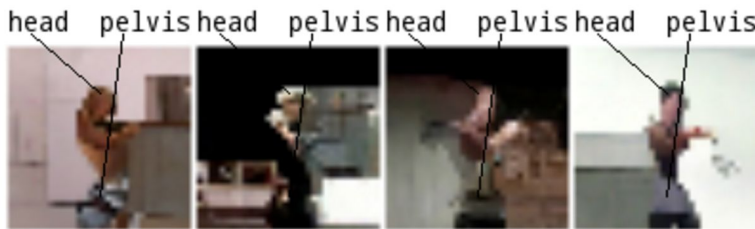

a man in a blue shirt and white cargo shorts holds up a carpeting tool.

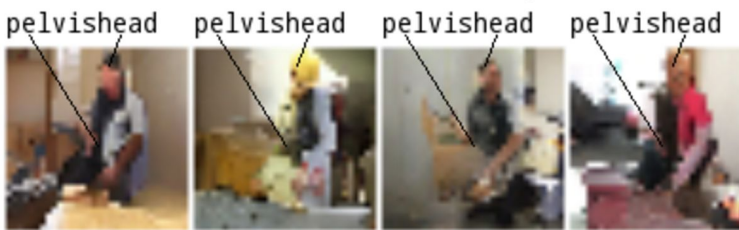

a man in khaki fishing gear and waders has caught a fish

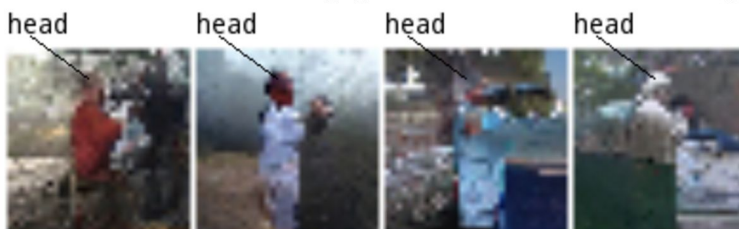

a man in a blue cap and grey shirt spray paints a house.

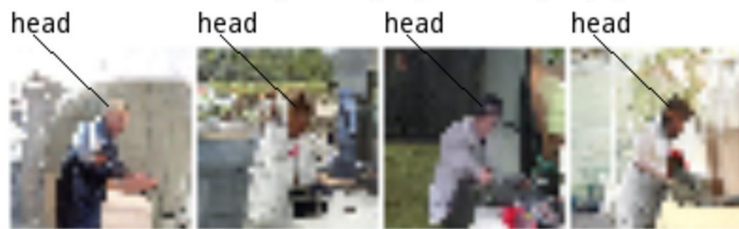

a person in a black jacket and neon yellow pants is skiing down a hill.

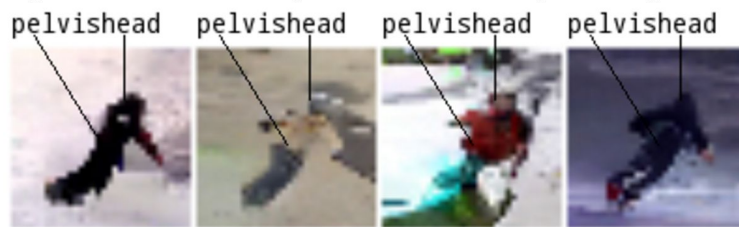

