# Peer review of "Generating Interpretable Images with Controllable Structure"

_ICLR 2017 — rejected_

[Official Review · AnonReviewer1 · rating 6 · confidence 3 · 17 Dec 2016]
**Interesting results, but comparisons seem lacking**

"First, it allows us to assess whether auto-regressive models are able to match the GAN results of Reed et al. (2016a)." Does it, though? Because the resolution is so bad. And resolution limitations aren't addressed until the second-to-last paragraph of the paper. And Figure 9 only shows 3 results (picked randomly? Picked to be favorable to PixelCNN?). That hardly seems conclusive.

The segmentation masks and keypoints are pretty strong input constraints. It's hard to tell how much coherent object and scene detail is emerging because the resolution is so low. For example, the cows in figure 5 look like color blobs, basically. Any color blob that follows an exact cow segmentation mask will look cow-like.

The amount of variation is impressive, though.

How can we assess how much the model is "replaying" training data? Figure 8 tries to get at this, but I wonder how much each of the "red birds", for instance, is mostly copied from a particular training example.

I'm unsatisfied with the answers to the pre-review questions. You didn't answer my questions. The paper would benefit from concrete numbers on training time / epochs and testing time. You didn't say why you can't make high resolution comparisons. Yes, it's slower at test time. Is it prohibitively slow? Or is it slightly inconvenient? There really aren't that many comparisons in the paper, anyway, so it if takes an hour to generate a result that doesn't seem prohibitive. 

To be clear about my biases: I don't think PixelCNN is "the right way" to do deep image generation. Texture synthesis methods used these causal neighborhoods with some success, but only because there wasn't a clear way to do the optimization more globally (Kwatra et al, Texture Optimization for Example-based Synthesis being one of the first alternatives). It seems simply incorrect to make hard decisions about what pixel values should be in one part of the image before synthesizing another part of the image (Another texture synthesis strategy to help fight back against this strict causality was coarse-to-fine synthesis. And I do see some deep image synthesis methods exploring that). It seems much more correct to have a deeper network and let all output pixels be conditioned on all other pixels (this conditioning will implicitly emerge at intermediate parts of the network). That said, I could be totally wrong, and the advantages stated in the paper could outweigh the disadvantages. But this paper doesn't feel very honest about the disadvantages.

Overall, I think the results are somewhat tantalizing, especially the ability to generate diverse outputs. But the resolution is extremely low, especially compared to the richness of the inputs. The network gets a lot of hand holding from rich inputs (it does at least learn to obey them). 

The deep image synthesis literature is moving very quickly. The field needs to move on from "proof of concept" papers (the first to show a particular result is possible) to more thorough comparisons. This paper has an opportunity to be a more in depth comparison, but it's not very deep in that regard. There isn't really an apples to apples comparison between PixelCNN and GAN nor is there a conclusion statement about why that is impossible. There isn't any large scale comparison, either qualitative or quantified by user studies, about the quality of the results.

[Official Review · AnonReviewer2 · rating 7 · confidence 3 · 19 Dec 2016 (modified: 20 Jan 2017)]
**review: worthwhile extension of PixelCNN capabilities**

This work focuses on conditional image synthesis in the autoregressive framework.  Based on PixelCNN, it trains models that condition on text as well as segmentation masks or keypoints.  Experiments show results for keypoint conditional synthesis on the CUB (birds) and MHP (human pose) dataset, and segmentation conditional synthesis on MS-COCO (objects).  This extension to keypoint/segment conditioning is the primary contribution over existing PixelCNN work.  Qualitative comparison is made to GAN approaches for synthesis.

Pros:
(1) The paper demonstrates additional capabilities for image generation in the autoregressive framework, suggesting that it can keep pace with the latest capabilities of GANs.
(2) Qualitative comparison in Figure 9 suggests that PixelCNN and GAN-based methods may make different kinds of mistakes, with PixelCNN being more robust against introducing artifacts.
(3) Some effort is put forth to establish quantitative evaluation in terms of log-likelihoods (Table 1).

Cons:
(1) Comparison to other work is difficult and limited to qualitative results.  The qualitative results can still be somewhat difficult to interpret.  I believe supplementary material or an appendix with many additional examples could partially alleviate this problem.
(2) The extension of PixelCNN to conditioning on additional data is fairly straightforward.  This is a solid engineering contribution, but not a surprising new concept.

[Official Review · AnonReviewer3 · rating 5 · confidence 3 · 20 Dec 2016]
**Conditional PixelCNN**

This paper proposes an extension of PixelCNN method that can be conditioned on text and spatially-structured constraints (segmentation / keypoints). It is similar to Reed 2016a except the extension is built on top of PixelCNN instead of GAN. After reading the author's comment, I realized the argument is not that conditional PixelCNN is much better than conditional GAN. I think the authors can add more discussions about pros and cons of each model in the paper. I agree with the other reviewer that some analysis of training and generation time would be helpful. I understand it takes O(N) instead of O(1) for PixelCNN method to do sampling, but is that the main reason that the experiments can only be conducted in low resolution (32 x 32)? I also think since there are not quantitative comparisons, it makes more sense to show more visualizations than 3 examples. Overall, the generated results look reasonably good and have enough diversity. The color mistake is an issue where the author has provided some explanations in the comment. I would say the technical novelty is incremental since the extension is straightforward and similar to previous work. I lean toward accepting because it is very relevant to ICLR community and it provides a good opportunity for future investigation and comparison between different deep image synthesis methods.

[Author Response · Scott Reed · 18 Jan 2017]
**Rebuttal, addition of 64x64 and 128x128 samples**

We thank all reviewers for their detailed feedback, and note that all reviewers recommend the paper for acceptance. Based on reviewer feedback about image resolution, we trained a 64x64 and 128x128 version of the model on the CUB dataset, results of which can be seen at sites.google.com/view/iclr2017figures. These and additional higher resolution results will be added to the revised paper. Since low-resolution was one of the main drawbacks to the paper according to the reviews, we hope that this can be reflected in an improved score.

We posted an updated version of the paper to OpenReview with an important correction to the caption of table 1: likelihoods are in *nats* per dim, not bits.

Below we respond to each review individually.

AR1:

The time required for sampling is the main constraint on generating higher-resolution samples. However, we have been able to train some higher-resolution models in time for the rebuttal (see sites.google.com/view/iclr2017figures for some results). We agree that adding many more examples for comparison would help; these will be added in the upcoming version.  Please see the reply to AnonReviewer 3 for more precise details on timing and the experimental setup.

Please see other replies regarding comparison of GANs to pixelCNNs. In short, there are trade-offs between these two. We accentuate the trade-offs in the paper, in the hope that researchers will then know what are the key problems of each approach, and focus on developing solutions to those problems. A quantitative comparison is problematic because GANs don’t provide us with likelihoods. We can however include more samples. To this extent we will add more high-resolution samples, including the ones already provided via the link above.

AR2:

We appreciate your suggestion of adding more results to the appendix for the final version, or even better create a website where users can explore the generated images by both approaches. The figures in the paper provide a reasonable depiction of the trade-offs between existing GANs and pixelCNNs, but we agree adding more comparisons will help.

AR3:

Matching GANs:
In the paper we demonstrate that autoregressive models can do text- and location-conditional image generation; although as the reviewer points out, the resolution is much lower so “match” is not the right word; GANs and auto-regressive models have complementary strengths and weaknesses. We are happy to add a more thorough discussion of these issues earlier in the paper, rather than at the end. Figure 9 queries were from a figure of positive results in the paper to which we compare - so presumably favourable to GAN - but we agree that many more comparisons are needed to study the different types of mistakes each method makes.

Replaying training data:
One way to check for this is to compare likelihoods for the training set and held-out sets of data. In our case we did not observe significant overfitting, so copying seems unlikely. We also observe significant diversity of samples even with fixed text and structure constraints. However, as noted in the GAN paper to which we compare, even if the model had largely memorized the (text,location,image) training data, it could still produce novel images by conditioning on novel combinations of (text,location), or in general the combinatorial space of all its conditioning variables.

More implementation details:
Currently the paper says “We used RMSprop with a learning rate schedule starting at 1e-4 and decaying to 1e-5, trained for 200k steps with batch size of 128”. Additional details: The number of epochs varies by dataset - more for CUB because it is smaller, fewer for MS-COCO. Training took about 4 days and sampling at 32x32 resolution took about 2 minutes per image with batch size of 30. Sampling 64x64 took about 16 minutes per image, and 128x128 took about 2 hours. (However, note that sampling time is highly implementation dependent, and we used only the most naive approach in this paper).

High-resolution comparisons:
We will add further hi-res comparisons in the revised paper.

Autoregressive approach:
I (first author) also have a bias toward GANs, having written several papers using them. However, I also think autoregressive approaches have complementary benefits compared to GANs - stable, easy to train, do not overfit to a few modes, best available image density estimators, etc - and are worth further developing. Also, autoregressive and adversarial approaches could be naturally combined; e.g. as likelihood and posterior models in PPGNs (

[Final Decision · Program Chairs · 06 Feb 2017]
**ICLR committee final decision**

The paper extends PixelCNN to do text and location conditional image generation. The reviewers praise the diversity of the generated samples, which seems like the strongest result of the paper. On the other hand, they are concerned with their low resolution. The authors made an effort of showing a few high-resolution samples in the rebuttal, which indeed look better. Two reviewers mention that the work with respect to PixelCNN is very incremental, and the AC agrees. Overall, this paper is very borderline. While all reviewers became slightly more positive, none was particularly swayed. The paper will make a nice workshop contribution.